# A Snapshot of the Most Recent Transthyretin Stabilizers

**DOI:** 10.3390/ijms25189969

**Published:** 2024-09-16

**Authors:** Carlo Marotta, Lidia Ciccone, Elisabetta Orlandini, Armando Rossello, Susanna Nencetti

**Affiliations:** 1Department of Pharmacy, University of Pisa, Via Bonanno 6, 56126 Pisa, Italy; carlo.marotta@phd.unipi.it (C.M.); armando.rossello@unipi.it (A.R.); susanna.nencetti@unipi.it (S.N.); 2Department of Earth Sciences, University of Pisa, Via Santa Maria 53-55, 56100 Pisa, Italy; elisabetta.orlandini@unipi.it

**Keywords:** transthyretin, TTR, stabilizers, binding affinity, crystallographic studies, activity, structure–activity relationship studies, SAR, synthetic compounds

## Abstract

In recent years, several strategies have been developed for the treatment of transthyretin-related amyloidosis, whose complex clinical manifestations involve cardiomyopathy and polyneuropathy. In view of this, transthyretin stabilizers represent a major cornerstone in treatment thanks to the introduction of tafamidis into therapy and the entry of acoramidis into clinical trials. However, the clinical treatment of transthyretin-related amyloidosis still presents several challenges, urging the development of new and improved therapeutics. Bearing this in mind, in this paper, the most promising among the recently published transthyretin stabilizers were reviewed. Their activity was described to provide some insights into their clinical potential, and crystallographic data were provided to explain their modes of action. Finally, structure–activity relationship studies were performed to give some guidance to future researchers aiming to synthesize new transthyretin stabilizers. Interestingly, some new details emerged with respect to the previously known general rules that guided the design of new compounds.

## 1. Introduction

In this article, the latest advancements (from 2020 onward) in the treatment of amyloidosis associated with the misfolding and fibrillization of the transthyretin (TTR) protein will be reviewed. In particular, this review will focus on synthetic TTR stabilizers, with the aim of providing an overview of the most recent and promising scaffolds that have been reported in this field. To assist future researchers with the design of new and improved TTR stabilizers, structure–activity relationship (SAR) schemes will be provided when possible. For an overview of the advancements published before the period covered by this paper, readers can view the following papers [1,2,3].

### 1.1. Physiologic Role of TTR

TTR, also called prealbumin, is a circulating plasma protein constituted by four identical monomers (A, A’, B, B’), assembled through a 2-fold axis of symmetry forming a homotetramer. From a structural point of view, each monomer (127 amino acids) is characterized by eight antiparallel β-strands (DAGH and CBEF) organized into two β-sheets that form a β-barrel, with one α-helix between the β-strands E and F (formerly named the EF-helix). TTR can be considered to be a dimer of dimers where dimerization takes place between the strands H and F of each monomer (Figure 1a,b). At the dimer–dimer interface, a channel arises which crosses the tetramer, forming two symmetric thyroxine binding pockets (T_4_BPs) [4]. Two molecules of T_4_ bind the TTR tetramer as follows: one at the interface of the monomers A/A’ and another between the B/B’ subunities, with an occupancy of 0.5, respectively. Each T_4_BP is characterized by three symmetric subsites which accommodate the four iodine atoms of T_4_, named halogen-binding pockets (HBPs); from the outside to the inside, they are labeled HBP1, 1′, 2, 2′, 3, and 3′. HBP3 is located between the side chains of Ser117, Thr119, Ala108, and Leu110, while HBP2 is characterized by the side chains of Leu110, Ala108, and Ala109, and HPB1 is contoured by Glu54 and Lys15. As depicted in Figure 1c, the crystal structure of T_4_ in complex with TTR (pdb id 2ROX) shows that the hormone extends deep into the BPs, with the phenolic group pointing towards Ser117 and Thr119 (HBP3), while the alanyl moiety is oriented against the entrance of HBP1 (Glu54 and Lys15). This kind of orientation is conventionally defined as the *forward* binding mode [5]. The T_4_ iodine atoms establish the main interactions with Ala109 and Leu110 (HBP2/3). Interestingly, a single point mutation of Leu110 with an Ala decreases the hydrophobicity of HBP3/HBP2 and, thus, promotes the fast dissociation of the tetramer into monomers [6].

Although the liver is the main producer of TTR in the body, other sites for its synthesis include the eyes (specifically, the retinal pigment epithelial cells), the choroid plexuses of the brain, and the α-cells of the islets of Langerhans [7,8,9,10,11,12,13]. This protein has a half-life of around 48 h in the blood, after which it undergoes degradation by the skin, liver, and muscles [14,15]. 

**Figure 1 ijms-25-09969-f001:**
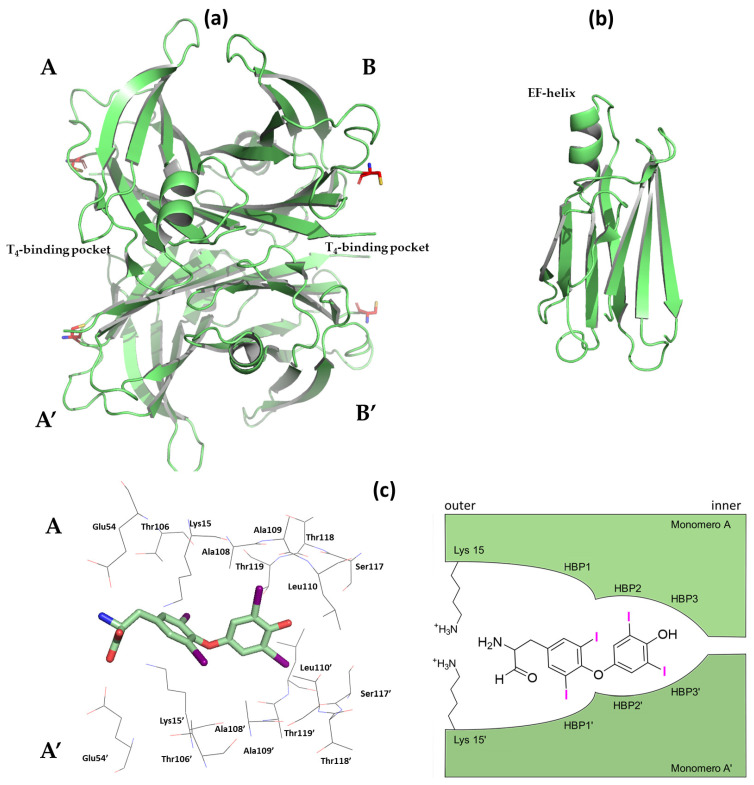
Graphical representation of the human TTR crystal structure. All the structural figures were drawn by author using the Protein Data Bank (PDB) id and PyMol program scripts [16,17]. (**a**) TTR tetramer, Cys10 in red. (**b**) TTR monomer. (**c**) Left TTR-T4 binding mode (pdb id: 2ROX), right scheme of halogen-binding pockets (HBPs).

The physiologic role of TTR is implicitly described through its acronym. Indeed, it is responsible for the transport of both thyroxine (T_4_) and retinol (thus, T-T-R, as in Transport–Thyroxine–Retinol). In this respect, TTR is recognized as the second major transporter of T_4_ in the plasma and as the first in the cerebrospinal fluid (CSF) [18]. Furthermore, by binding to the retinol-binding protein (RBP), it forms a complex which acts as a transporter for retinol (also known as vitamin A) [19]. TTR’s participation in this complex is important, as it is generally accepted that it avoids the filtration of the RBP–retinol complex by the kidney [20,21,22]. Also noteworthy is that, besides these activities, TTR is also endowed with proteolytic activity toward several proteins, such as the Aβ peptide, which has great implications in Alzheimer’s disease [23,24,25,26,27].

### 1.2. Pathologic Role of TTR

In contrast with its physiological role, TTR is also associated with a set of disorders known as “transthyretin amyloidosis” (ATTR). Indeed, TTR can undergo a misfolding process which results in the deposition of amyloidogenic fibrils in the body [28]. This process occurs because of the intrinsic instability of the TTR protein, which can be due to either aging or single point mutations in the gene encoding it [28,29]. The first situation (namely, aging) results in the deposition of wild-type TTR (wtTTR) fibrils, causing a condition known as “wild-type transthyretin amyloidosis” (ATTRwt) (which was previously called “senile systemic amyloidosis”) [28,30]. On the other hand, the latter causes the synthesis of less stable mutated TTR, which results in a condition known as “transthyretin amyloidosis variants” (ATTRvs), also referred to as “mutant” or “hereditary transthyretin amyloidosis” [29,30]. Because of these distinct characteristics, ATTRwt is considered to be an age-related disease, while ATTRvs is hereditary [28,31].

Although the process of amyloid fibril formation has not yet been completely understood, it is generally accepted that it involves the dissociation of the TTR tetramer into dimers and, subsequently, into monomers that lose their native structure and aggregate, thus forming insoluble deposits [32,33]. This process is promoted by the aforementioned single point mutations, which destabilize the structure of the TTR protein, but it can also take place in elderly patients, because the large amount of β-strands within the TTR’s structure make it inherently amyloidogenic [34]. 

It has been observed that the monomers’ structures differ between the TTR tetramer (native conformation) and the fibrils. This is due to the fact that the monomers have to undergo an unfolding process in order to be able to aggregate and form mature fibrils. In this regard, it has been reported that these monomers are 10^5^ times less stable than the tetramer to the unfolding process. Therefore, since TTR is a protein endowed with a remarkable stability, its dissociation into less stable monomers is a mandatory step for the fibrils’ formation [35,36,37,38]. Regarding the specific conditions driving the formation of these fibrils from wtTTR in vivo, several hypotheses have been proposed. For example, it has been postulated that it involves the acidic environment of lysosomes and endosomes or proteolytic cleavage under mechanical stress by proteases (such as plasmin) [39,40,41]. Moreover, a relevant role has been attributed to metal dyshomeostasis because of the effect of bivalent metals, in particular on His88 and His90. In addition, the oxidative stress resulting from the action of reactive oxygen species (ROS) on Cys10 seems to play a role as well (Figure 1) [42,43,44].

In light of this, it is clear that molecules capable of stabilizing the TTR tetramer, thus preventing its dissociation and subsequent misfolding, are promising therapeutic agents for the treatment of ATTR. Usually, such stabilization occurs through their binding to TTR binding sites and, in this regard, it is worth mentioning that they can bind in a non-cooperative or cooperative way (the latter can be further divided into positive and negative cooperativity) [45,46,47]. Negative cooperativity is the most frequent binding mode for TTR ligands and it means that, after binding to the first TTR binding site, the affinity of the ligand for the second site decreases [48,49,50,51]. Therefore, ideally, ligands should either show positive cooperativity or non-cooperativity [48]. This is one of the challenges faced in the development of suitable TTR-stabilizing agents and it justifies the difficulties in finding suitable drug candidates. However, this field is still very promising, as demonstrated by the recent introduction of tafamidis (Vyndaqel; Pfizer, New York, Stati Uniti) into therapy [50], the first-in-class and only drug approved for the treatment of ATTRs [52,53,54]. However, due to its limitations (which are described in the next paragraphs), there is still a great need for new therapeutics. In view of this, some interesting TTR stabilizers have recently been reported in the literature, and the most promising among them are described in detail in the sections below. 

### 1.3. Clinical Manifestations of ATTR: Amyloidotic Cardiomyopathy and Polyneuropathy

As already mentioned in the previous paragraph, ATTR involves the accumulation of amyloidogenic fibrils in several different districts of the body [28]. If the deposits are concentrated in the heart, the patient develops a disease called “amyloidotic cardiomyopathy” (AC), while, if they are located in the nerves, so-called “amyloidotic polyneuropathy” (AP) arises [31,55]. In particular, ATTR-AC results from the deposition of fibrils formed either from wtTTR or mutated TTR and, therefore, ATTR-AC can be either ATTRwt or ATTRv [55,56].

ATTR-AC is a fatal disease in which misfolded TTR forms deposits in the myocardium, thus leading to a decline in cardiac functionality and, ultimately, to heart failure [31,55]. For several decades, ATTR-AC was considered to be a rare disorder, while, in recent years, thanks to great innovations in diagnostic imaging, an increasing incidence has been recorded [54,57,58]. Indeed, an epidemiological study reported that, in the UK, the incidence of ATTRwt amyloidosis increased from less than 3% in 1987–2009 to 14% in 2010–2015, finally reaching 25% in the period of 2015–2019 [59]. However, despite the advances in diagnostics, the ratio between the incidence and prevalence of wtATTR in the population remains unknown [59]. Usually, ATTRwt-AC’s symptoms tend to manifest in old age (50–80 years old), while those of ATTRv-AC can also appear early in life [53,55,60]. To date, more than 150 mutations have been reported to cause ATTRv-AC (the most common are Thr60Ala, Leu111Met, Val122Ile, and Ile68Leu), while ATTRwt-AC has been described to manifest mainly in men rather than women [60,61,62,63]. Additionally, recent studies have ascribed the low life expectancy (from around 2 to 3.5 years) of patients suffering from AC to the difficulty in making a correct diagnosis of this disease (indeed, the average reported time for its diagnosis is 3.6 years) [55,64]. This depends on its non-specific symptoms that can lead to mistaking it for other pathologies. In addition, invasive analyses, such as endomyocardial biopsy, are needed to recognize it with certainty. Overall, this has been reported to result in incorrect treatment in some cases [55]. Also noteworthy, the incidence of this disease has been re-evaluated in recent years, and it is now no longer regarded as rare. Fortunately, thanks to many scientific breakthroughs, therapeutic options have increased, and it is now a more clinically manageable condition [65].

As for AP, the corresponding fibrils form because of single point mutations in the gene encoding for TTR, which results in a weakening of protein stability. In particular, Val30Met is the most common mutation in AP, but other frequent ones are Gly47Glu, Thr60Ala, Ser77Tyr, and Glu89Gln [63,66]. However, many other mutations have been described, and most of them lead to illness [67]. Also noteworthy, while progressive peripheral neuropathy is a common symptom, there are also others that are specific to this condition, depending on the time of disease onset. For instance, gastrointestinal disorders and autonomic neuropathy are characteristic of early-onset patients bearing the Val30Met mutation, while heart disorders and motor neuropathy usually develop in late-onset cases [68]. Without appropriate treatment, this disease can lead to death in 10 years due to complications in the autonomic and peripheral nervous systems [69,70]. Similarly to AC, AP was considered to be an incurable condition in the past, but nowadays, thanks to new available therapies, it has become more clinically manageable [71]. 

In the next paragraph, the currently available therapeutic strategies for ATTR will be described.

### 1.4. Current Therapeutic Strategies and New Frontier Approaches

From a historical point of view, the first therapy that was devised for the treatment of ATTR was liver transplantation. However, many new strategies have been developed over the years, resulting in an improvement in the quality of life of patients. These new therapies include gene silencers, from small RNA-interfering drugs (such as patisiran and vutrisiran) to antisense oligonucleotides (like inotersen and eplontersen) and stabilizers of the TTR tetramer (namely tafamidis, diflunisal, tolcapone, and acoramidis) (Figure 2). 

Other promising approaches, such as CRISPR-CAS9 gene editing and fibril-disrupting drugs, are currently under development (Figure 3). In this paragraph, a brief description of these therapies will be given, with the aim of providing an overview of the current advances in the clinical treatment of ATTR.

Liver transplantation has been employed for a long time in the treatment of ATTRv, because, as already said, TTR is mostly produced in the liver. Therefore, through this surgical operation, the circulating TTR becomes wild-type (instead of mutated). Nonetheless, this approach is non-resolutive, because even after liver replacement, wt-TTR-derived amyloid fibrils (originating from the new liver) continue to form, forming deposits in the heart that result in the aforementioned pathologies [56]. To overcome this problem, new therapies (described below) have emerged over the years. 

Another therapeutic option involves the use of small interfering RNA drugs to induce the degradation of the mRNA coding for the production of the TTR protein, thus decreasing its plasma levels [30,72]. Given that TTR is responsible for the transporting of retinol, this treatment needs to be associated with vitamin A supplementation [30]. This class of drugs includes two clinically approved therapeutics, as follows: patisiran and vutrisiran. Patisiran is conveyed in lipid nanoparticles for a selective target of the mRNA inside hepatocytes [30,72]. It is currently being used for the treatment of ATTRv-AP, but it has also shown efficacy for the treatment of ATTR-AC [30,72,73]. It must be administered every 3 weeks by intravenous infusion and must be associated with premedication to prevent infusion-related adverse reactions related to its formulation in lipid nanoparticles [30,74]. Similar to patisiran, vutrisiran is approved for the treatment of ATTRv-AP and is capable of selective delivery to hepatocytes [30,74]. This last feature derives from the fact that it is tethered to N-acetyl galactosamine, a moiety endowed with a strong affinity for the asialoglycoprotein receptor, which is highly expressed in the aforementioned cells [30,74]. Compared with patisiran, vutrisiran is more manageable, because it can be administered every 3 months by subcutaneous infusion and it does not need premedication [30,74]. 

Similar to RNA-interfering drugs, antisense oligonucleotides bind to the mRNA coding for TTR production and lead to its degradation, thus decreasing the protein expression in cells [75,76]. For similar reasons to the previous class, these drugs should also be associated with vitamin A supplementation [30]. Within this class of drugs, there are inotersen and eplontersen. These two therapeutics share the same nucleotide sequence, but the latter is tethered to a triantennary *N*-acetylgalactosamine portion that allows it to undergo receptor-mediated uptake by hepatocyte cells. This mechanism enhances the drug’s potency and allows for using a lower dosage and less frequent administration regimen with respect to inotersen [77]. Both inotersen and eplontersen have received approval by the FDA for the treatment of ATTRv-AP, but the latter has been reported to have fewer side effects [30,78].

However, another strategy that does not require life-long drug administration is also under development, namely the CRISPR-CAS9 gene editing approach [30,79]. This strategy involves the introduction of breaks into the double-strand DNA and its consequent editing for a selective knock-down of TTR expression [80,81,82,83,84]. This treatment is particularly suited for diseases like ATTR, because TTR production is mainly localized in the liver and, therefore, only this tissue should be targeted by this modification. In addition, since the physiological role of this protein is limited, its knock-down should have few side effects [22,85]. 

Another attractive therapeutic option involves the use of monoclonal antibodies to target the already formed fibrillar deposits. The advantage of this approach is the possibility of reversing the disease progression and regaining organ functionality [86,87]. However, other already clinically approved drugs have also been repurposed for the disruption of fibrils, namely doxycycline and tauroursodeoxycholic acid (TUDCA), which have also been studied in combination therapy [88,89,90]. Since these strategies are beyond the scope of this review, they will not be described any further. However, if the reader desires to learn more about them, the following reviews are available [30,36,91].

Another strategy that has received much attention from the scientific community relies on the stabilization of the TTR tetramer. Drugs with such mechanisms of action have the purpose of preventing the dissociation and consequent misfolding of the TTR tetramer, thus avoiding fibril formation. As of today, there are three TTR stabilizers available in clinical practice, namely tafamidis, diflunisal, and tolcapone (Figure 2), whose mechanisms of action involve binding to the T_4_ binding sites of TTR [46,92,93,94]. Tafamidis has received approval for the treatment of both ATTR-AC and ATTR-AP, but, although it represents a breakthrough for the treatment of amyloidosis, it is not resolutive [95,96,97]. In fact, despite its efficacy and safety, patients must remain in treatment with this drug for life which, due to its high cost, is inconvenient [98,99]. However, its clinical effectiveness also suffers from some limitations, because it has been reported to be ineffective in around 30% of patients [100]. An alternative is represented by diflunisal, a nonsteroidal anti-inflammatory drug (NSAID) whose cost is considerably lower than that of tafamidis. However, diflunisal is burdened with many side effects, including the gastrointestinal symptoms typical of the NSAID class, prompting scientists to look for other compounds with a better pharmacological profile [30]. This intense research led to the repurposing of tolcapone, which binds to TTR with the same mechanism as tafamidis and diflunisal, thus stabilizing its tetrameric conformation [92]. In particular, it has been reported to have an even higher affinity toward recombinant wtTTR than tafamidis, leading it to be considered as a promising candidate for the treatment of ATTR-AP in clinical trials [92,101]. Currently, it is being used off-label for the treatment of ATTR-AC [94]. However, this drug has many drawbacks as well, ranging from its hepatic side effects to its short half-life [102,103,104,105,106]. 

Notably, there are also other brand new drugs currently under investigation for the treatment of ATTR. One such example is acoramidis (previously known as AG-10) (Figure 2), which is also a stabilizer of the TTR tetramer, although its mechanism of action differs from that of the previous drugs [30]. In particular, by binding to TTR, acoramidis mimics the effect of the T119M mutation, which grants a remarkable stability to the protein (in particular, it appears to increase its stability against dissociation by more than 33-fold compared with wtTTR) [46,107,108,109,110,111]. In light of this, it is clear that acoramidis is a very promising drug for the treatment of ATTR and, as a matter of fact, it is currently under investigation in clinical trials. 

In light of this, it is evident that developing new and more effective TTR stabilizers is important, both to widen the arsenal of drugs at our disposal and to overcome the limitations of those currently employed in therapy. The most recent compounds developed for this purpose are described in the next sections.

## 2. New TTR Stabilizers: From Drug Repurposing to New Promising Scaffolds

Drug repurposing has emerged as an advantageous strategy for reducing the time and cost of clinical trials with respect to the traditional drug discovery approach. This section will describe some drugs that have already been approved for the treatment of other diseases and have been explored for their TTR inhibitory activity. Moreover, new compounds derived from the chemical modification of these derivatives or other well-known TTR stabilizers will be presented. Also noteworthy, although several molecules have been reported over the years, only the most promising ones are described, with the aim of providing some guidance for future research directions. To achieve this purpose, when possible, SAR schemes are also drawn. 

### 2.1. The Most Recent Developments in the Optimization of Tolcapone’s Structure

As already said in the previous sections, tolcapone is a very efficient stabilizer of TTR, even more effective than tafamidis, and it is currently being used off-label for the treatment of ATTR-AC [92,94,101]. However, tolcapone presents several problems, ranging from its hepatotoxicity to its short half-life. Both these problems seem to derive from its metabolic pathway, which involves a glucuronidation of the 3-OH group [102,103,104,105,106]. With the aim of overcoming these limitations, Poonsiri et al. recently reported three new compounds with a methoxy group in position 3, namely **1**, **2**, and **3** (Figure 4) which, thanks to this modification, are less susceptible to the glucuronidation reaction (compound **1** is one of the metabolites of tolcapone) [102,106,112]. Thanks to these structural modifications, these compounds’ safety profiles were improved with respect to that of tolcapone [102]. However, since the *p*-CH_3_ phenyl ring of tolcapone does not create significant enthalpic interactions with the TTR binding site, modifications to that portion were also made in **2** and **3** to improve the drugs’ efficacy [92,102]. The TTR stabilization capability of these compounds was assessed by incubating them with human plasma and subsequently determining the abundance of the TTR monomer. The results revealed that **3** is the most promising among the three compounds, while **2** has an activity comparable to that of **1**, which is nonetheless still higher than that of tolcapone. According to the authors, this suggests the importance of the presence and location of the two –CH_3_ groups on the second phenyl ring [102].

The dissociation constant (K_d_) and the enthalpy of the interaction (ΔH) of these compounds with TTR were also evaluated. From the results, the authors concluded that the methylation of the 3-OH of these compounds weakened their binding strength with respect to tolcapone. On the other hand, the introduction of the second –CH_3_ into compound **3** enhanced both its affinity and enthalpy of interaction with respect to **1**. However, the disposition of the –CH_3_ moieties, as in compound **2,** worsened its affinity while enhancing the ΔH of the binding with respect to **3** (Table 1). The results of the thermodynamic studies correlated with the conclusions drawn from the studies on the stabilization of TTR, because compounds with a more negative ΔH were also capable of better stabilizing TTR. Overall, all these compounds were shown to bind better to TTR than tafamidis, making them promising candidates for the treatment of ATTR. In addition, all of them were also proven to be capable of crossing the blood–brain barrier more efficiently than tolcapone, which is a useful feature for preventing amyloid aggregation in the central nervous system [102].

From a structural point of view, crystallographic studies showed that compounds **1**–**3** bind TTR in a similar way to tolcapone, occupying both T_4_BPs in the “forward binding mode”, that is, with the polar moiety located in the outermost part of the channel (HBP1, 1′) (Figure 5a–d) [102,112]. In detail, the superposition of compounds **1**–**3** with tolcapone (Figure 5b) clearly shows that the polar ring (which is functionalized with hydroxy, methoxy, and nitro groups) is oriented towards Lys15, while the second ring (bearing the methyl or dimethyl substituents) is located deeper in the inner cavity. Interestingly, although compounds **1** and **2** bind to the T_4_BPs in the same way, a slight difference can be detected around the second phenyl ring. Indeed, in order to better accommodate the two methyl portions of compound **2** in HBP3 and HBP3′, its 3,5-dimethyl-phenil ring results in being shifted with respect to the corresponding ring of **1** (Figure 5c) [102,112]. Regarding compound **3**, its binding mode is identical to that of tolcapone, as illustrated in Figure 5d. 

Therefore, among these three compounds, **3** showed the same K_d_ and ΔH values as tolcapone, suggesting a high selectivity for TTR. In addition, X-ray analysis confirmed that **3** and tolcapone bind to TTR in the same way. In light of all these data, the authors concluded that compound **3** is the most promising, and recommended it for further studies [102]. 

Interestingly, the methylation of the 3-OH group was also explored by other researchers for the synthesis of compound **5** (Figure 4) [113]. This compound was designed as a development of compound **4** (Figure 4), which was previously reported to have a strong affinity for TTR (Table 1), but whose clinical use is limited by pharmacokinetic issues (specifically, a short plasma half-life and low bioavailability) [48,113]. Compared with the previous series (**1**–**3**), compound **4** features halogen on its second phenyl ring, which was introduced because it is capable of establishing stronger interactions with the TTR cavity compared with the para-methyl group of tolcapone [48,116]. This, together with the adjacent 5-OH group, allows **4** to have more interactions with the TTR cavity than tafamidis and tolcapone, thus justifying its improved features with respect to these drugs. As a matter of fact, it has been reported that compound **4** has a higher binding affinity for TTR (Table 1) and is a stronger inhibitor of the TTR dissociation in human plasma than both tafamidis and tolcapone. In addition, it has also been reported to be an efficient kinetic stabilizer and to effectively inhibit the amyloid fibril formation at an acidic pH [48]. 

The X-ray analysis of compound **4** in complex with wt-TTR confirms that, similar to tolcapone, this compound binds the T_4_BPs in the “forward binding mode”. The 3,4-dihydroxy-5-nitrophenyl ring of **4** is oriented towards the outer cavity, contoured by the hydrophobic residues of HBP2, while the 3-F substituted ring is located deeper in the inner binding cavity (Figure 6a). Also noteworthy, the fluorine atom of **4** allows it to establish additional interactions with the amino acid side chains of HBP3 with respect to tolcapone, which justifies its higher binding affinity for TTR than tolcapone [48]. 

Given the promising activity of **4**, researchers have tried to optimize its structure with the aim of overcoming the aforementioned pharmacokinetic issues. In view of this, since the 3-OH methylated metabolite of tolcapone (namely, compound **1**) has been reported to have a longer plasma half-life than tolcapone and be capable of stabilizing TTR [106,112], the same modification was applied to **4**, thus leading to compound **5** (Figure 4) [113]. 

Cytotoxicity studies on human cervical carcinoma (HeLa), human hepatoblastoma (HepG2), and normal human fibroblast (MRC-5) cells showed that compound **5** has a satisfying safety profile. In addition, studies on CD-1 male mice revealed that compound **5** has better pharmacokinetic features than tolcapone. Moreover, the capability of **5** to inhibit urea-induced wt-, V30M-, and V122I-TTR tetramer dissociation was evaluated, and the results showed that it has similar activity to tolcapone. Furthermore, **5** was also proven to be either more than (in wt- and V30M-TTR) or equally efficient (in V122I-TTR) to tolcapone in decreasing the acid-induced aggregation of TTR. Further experiments showed that **5** binds to wt-, V30M-, and V122I-TTR with a higher affinity than tolcapone (and, in the case of V30M- and V122I-TTR, tafamidis as well) and that these interactions are non-cooperative. The binding parameters of **5** are reported in Table 1. According to the authors, these values suggest that the binding of **5** to TTR involves the formation of non-covalent interactions [113].

In order to investigate its binding mode at the atomic level, the crystal structures of compound **5** in complex with wt-TTR, V30M-TTR, and V122I-TTR were solved. The data showed that, similar to the previous results, compound **5** interacts with the T_4_BPs in the “forward binding mode” (Figure 6b). No relevant differences were found between the binding modes of compound **5** with wt-TTR and with the studied mutants (Figure 6b). In addition, similar to compound **4**, the fluorine atom of compound **5** establishes strong interactions with HBP3 of the T_4_BPs, allowing the 3-fluoro-5-hydroxyphenyl ring to orient deeper in the inner binding pocket than the corresponding ring of tolcapone [113]. 

To better characterize its activity profile, the ability of **5** to compete with T_4_ for binding to TTR was evaluated in plasma samples from wt- and V30M-TTR patients. The results revealed that **5** displaces T_4_ from its binding to TTR more efficiently than tolcapone. Moreover, **5** was also shown to stabilize wt- and V30M-TTR more efficiently than tolcapone in experiments performed on human plasma samples. The authors hypothesized that one of the reasons for the better performance of **5** with respect to tolcapone could reside in its capability to establish additional bonds with the TTR cavity. In light of this, the authors proposed **5** for the treatment of ATTR, especially for V30M-ATTRv-AP [113].

Overall, the results obtained from these studies on compounds **1**–**5** strongly suggest that new tolcapone-derived molecules might benefit both in terms of their safety profile and pharmacokinetics from the methylation of the 3-OH moiety of the first phenyl ring (Figure 7). Moreover, as already said, the p-CH_3_ phenyl ring of tolcapone does not create significant enthalpic interactions with the TTR binding site and, therefore, appropriate modifications to this ring could be useful for improving the drug’s efficacy [92]. In view of this, the simultaneous functionalization of the 2 and 4 or 3 and 5 positions of this ring would allow for the establishment of additional interactions with the TTR cavity, thus improving the TTR-stabilizing capability of these compounds. Regarding the 2,4 substituents, methyl moieties were proven to lead to satisfactory results, while for 3,5 functionalization, the available data suggest that the presence of halogen at position 3 (such as F) and an OH group at position 5 is beneficial (Figure 7). On the other hand, it seems that the remaining portions of the tolcapone structure should be kept unmodified for the optimal binding. 

### 2.2. Beyond Tolcapone: New Scaffolds from Anthelmintic Drugs

In the search for new treatments for ATTR, Yokoyama et al. recently tested many clinically approved drugs for their TTR-stabilizing activity. Among them, the most promising were bithionol (**6**) (a drug used in the past for the treatment of diphyllobothriasis infections [117,118,119,120,121]) and triclabendazole (**7**) (which is employed for the treatment of human fascioliasis [122]) (Figure 8) [123]. 

Yokoyama et al. performed a competitive binding assay using 8-Anilinonaphthalene-1-sulfonic acid (ANS) on both of these compounds with V30M-TTR. The results showed that **7** has a binding affinity similar to that of tafamidis, while that of **6** is even higher. In the case of **6**, these results were further confirmed by a tryptophan intrinsic fluorescence assay performed on V30M-TTR, which showed that this drug can indeed bind to the TTR tetramer (**7** could not be tested because of interference problems). Nonetheless, the potential of both these drugs for ATTR treatment was further explored. Indeed, they were both proven to have a similar capability to tafamidis, diflunisal, and tolcapone for inhibiting acid-induced amyloid fibril formation by V30M-TTR. Subsequent tests performed on V30M-TTR revealed that this inhibitory activity derived from their capability to stabilize the TTR tetramer. In addition, tafamidis and **7** were ineffective in inhibiting the methanol-induced aggregation of wt-TTR, while tolcapone was only moderately active. On the other hand, **6** demonstrated a high activity and this, according to the authors, might depend on its strong affinity for TTR and its weak negative cooperativity in binding with it. To further clarify this point, the affinity of **6** for the TTR binding site of V30M-TTR was evaluated through ITC (K_d_ = 0.062 ± 0.013 μM, ΔH = −6.0 ± 0.29 kcal mol^−1^). By comparing these values with those of tafamidis, diflunisal, and tolcapone, the authors concluded that **6** binds with a stronger affinity than the others to V30M-TTR [123].

The X-ray analysis confirmed that compounds **6** and **7** are able to bind to TTR (Figure 9). Regarding compound **6**, the authors decided to describe its binding mode with wt-TTR, because the ligand density for the TTR-V30M crystal complex was not clear. In this regard, compound **6** interacts with the T_4_BPs by fully occupying the three symmetric hydrophobic depressions (namely, the HBPs) with its chlorine atoms (Figure 9a) [123]. As for compound **7**, its crystal structure in complex with TTR-V30M shows that the 2,3-dichlorophenyl ring is located deeper in HBP3/3′, with the third chlorine atom (the one in the 5′ position) being oriented in HBP2 (Figure 9b). Finally, the thiomethyl tail of **7** points towards the entrance of the channel, around HBP1 (Figure 9b) [123].

Overall, among the studied compounds, the authors considered **6** as the most promising. Indeed, this study is interesting, because drug **6**′s structure does not present the typical carboxy moiety of most TTR stabilizers (which is responsible for the formation of polar interactions with the amino acid side chains of the TTR cavity). Therefore, compound **6** could represent a promising scaffold worthy of further developments for a series of new TTR stabilizers [50,123,124,125]. 

### 2.3. New Scaffolds Containing Bicyclic Aromatic Rings

Another clinically approved drug that has been studied for ATTR is benziodarone (**9**) (Figure 10), which is employed in chronic gout to decrease hyperuricemia [126]. This molecule was studied as a result of the research performed on benzbromarone (**8**) (Figure 10), a uricosuric drug that has previously been reported to be capable of stabilizing TTR with a similar efficiency to tafamidis and tolcapone by binding to its T_4_ binding site [127,128]. Starting from this scaffold, Mizuguchi et al. studied the potential of similar molecules for ATTR treatment, namely **9** and its 6-OH derivative (**10**) (Figure 10) [126]. In an acid-induced aggregation assay on V30M-TTR, both **9** and **10** showed strong activity, similar to that of tafamidis. Further experiments performed on V30M-TTR revealed that **9** and **10** have a comparable affinity for the T_4_ binding site of TTR. Moreover, their capability to selectively bind to TTR in human plasma confirms their potential for ATTR treatment [126].

Finally, a crystallographic analysis of these three promising compounds in complex with V30M-TTR confirmed their ability to bind with T_4_BPs. As shown in Figure 11a, the binding modes of derivatives **8** and **9** are identical. There is a perfect superposition between the halogen atoms (bromine for **8** and iodine for **9**), which point towards Lys15 at the entrance of the channel, while the benzofuran ring is located in the inner binding cavity [126].

A similar binding mode was found for derivative **10**, where iodine atoms occupy HBPs1/1′ in the outer cavity and the benzofuran ring is located in the inner binding pockets. The different orientation of the benzofuran moiety of compound **10** with respect to compounds **8** and **9** is due to the 6-hydroxy function that extends deeper into HBP3 and establishes polar interactions with the amino acid side chain (Figure 11b). 

Given these encouraging results, Mizuguchi et al. tried to modify the scaffold of **9** to obtain molecules with improved features. This work led to the synthesis of a new series of derivatives, and the most interesting ones were found to be compounds **11**–**15** (Figure 10) [129].

These compounds were proven to bind with a higher potency and selectivity to TTR in the plasma than tafamidis and **9** (with **13** and **15** giving better results than **11** and **14**). In addition, compounds **11**–**15** were found to be capable of inhibiting the acid-induced aggregation of V30M-TTR, with similar IC_50_ values to tafamidis. However, the authors pointed out that the best IC_50_ value they could possibly obtain in their assay was 5 μM (also noteworthy, the IC_50_ value of **9** and **11**–**15** was around 5 μM). To further characterize their activity on the TTR protein, the K_d_ values of compounds **11**–**15** were determined and are shown below in Table 2, along with the thermodynamic parameters of compound **8** for comparison. Notably, compounds **12**, **13**, and **15** were also shown to have a similar bioavailability to tafamidis in in vivo pharmacokinetic studies, making them interesting molecules [129].

From a structural point of view, the authors obtained the crystals of V30M-TTR in complex with **11** (pdb id 8WGS) and **14** (pdb id 8WGT) (Figure 12). The X-ray data show that compound **11** interacts with the T_4_BPs with two different poses. In the first binding mode, the hydroxy-diiodophenyl ring and the benzofuran moiety of **11** occupy the same position previously reported for **9,** with the only difference being due to the third iodine atom in position 4 (Figure 11a versus Figure 12a) [129]. 

In the second binding pose of compound **11,** the benzofuran group shows a different orientation, with the chlorine atom being perfectly located in HBP3. Interestingly, this second binding mode is analogous to that found for compound **14** (Figure 12b).

In the present paper, only the most promising compounds were discussed. However, the authors also synthesized other derivatives, whose study led to the development of the following SARs (Figure 13) [129]:(1)The benzofuran ring (R_1_ = O) can also be replaced by a 1H-indole ring (R_1_ = NH). Substitution with a benzothiophene (R_1_ = S) leads to a reduction in activity;(2)The binding potency to TTR in the plasma decreases if the R_2_ on benzofuran ring is functionalized with groups bulkier than an ethyl (the authors tried an isopropyl group);(3)The functionalization of positions 4 (R_3_) and 7 (R_6_) of the benzofuran ring with Cl atoms leads to more active compounds with respect to those bearing Cl atoms in positions 4 (R_3_) and 5 (R_4_) or 6 (R_5_) and 7 (R_6_);(4)A higher binding potency to TTR in the plasma is obtained if positions 4 (R_3_) and 7 (R_6_) of the benzofuran ring are functionalized with Cl atoms instead of F atoms;(5)Substituents should preferentially be placed in position 4 (R_3_) over positions 5 (R_4_), 6 (R_5_), or 7 (R_6_). In particular, the 4-Cl substitution (R_3_) on the benzofuran ring leads to a higher potency of binding to TTR with respect to 5-Cl (R_4_), 6-Cl (R_5_), and 7-Cl (R_6_);(6)The functionalization of R_4_ and R_5_ should be avoided (R_4_ and R_5_ = H);(7)For selective binding to TTR in the plasma, R_3_ should be functionalized either with I, Cl, Br, CF_3_, or CH_3_. Also noteworthy, the functionalization of R_3_ with F was studied, but it led to worse results than with I, Br, and Cl;(8)R_7_ and R_8_ should both be functionalized with halogen atoms. In particular, their functionalization with I atoms leads to more active compounds than with Br atoms;(9)For selective binding to TTR in the plasma, the C=O moiety is important.

These considerations are highly interesting, as they expand the arsenal of potential platforms for the development of new TTR stabilizers. Future studies might help to clarify the optimal type of aromatic rings to be employed and with which substituents they should be functionalized to obtain compounds with high activity.

## 3. Conclusions

ATTR is a complex disease with multiple clinical manifestations, ranging from AC to AP. Over the years, several strategies (such as gene silencers, CRISPR-CAS9 gene editing, and disruptors of fibrils) have been studied for its treatment. In view of this, stabilizers of the TTR tetramer have demonstrated particularly promising activity. Indeed, the recent introduction of tafamidis into therapy represents a major breakthrough for the treatment of TTR-related amyloidosis. Nonetheless, its drawbacks pose serious limitations for the clinical management of this disease. Therefore, researchers have tried to develop new and improved therapeutics, either by modifying the structures of already known TTR-binding agents or by studying the activity of new scaffolds. In the present paper, we described the most recent and interesting TTR stabilizers that have been reported in the literature, ranging from derivatives of tolcapone to new scaffolds derived from various sources. When possible, SAR studies were performed to provide some guidance to future researchers for the development of new compounds. 

Overall, some general rules can be drawn from this work for the design of new TTR stabilizers. Indeed, it was previously reported that TTR-binding agents usually have two aromatic rings with a linker [1]. This rule of thumb finds confirmation with the compounds reported in the present review. However, through their analysis and by comparing them with well-known TTR stabilizers such as tafamidis, diflunisal, and tolcapone, we can add that, normally, the first ring is a benzene, while the second one usually consists either of a second benzene ring or a bicyclic aromatic ring. In addition, the linker is usually very short, generally consisting of only one atom, or is even absent, as in the case of tafamidis and diflunisal. Regarding the substituents on these aromatic rings, they are generally not very bulky and usually consist of halogens, OH groups, or hydrophobic moieties. Finally, as reported in [123], it is reasonable to assume that the overall molecular weight of drugs should not be too high and that chain-shaped structures should be avoided, because the TTR-binding site cannot accommodate them. 

In conclusion, we believe that this and similar studies could be useful for the design of more efficient TTR stabilizers capable of overcoming the limitations of the drugs currently employed in clinical practice.

## Figures and Tables

**Figure 2 ijms-25-09969-f002:**
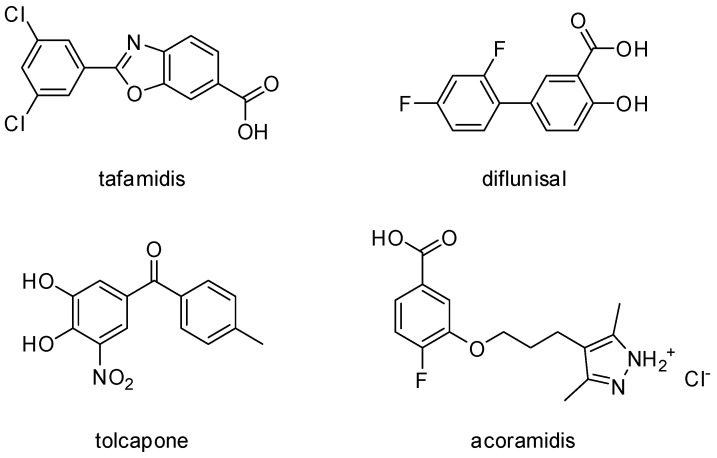
Chemical structures of tafamidis, diflunisal, tolcapone, and acoramidis.

**Figure 3 ijms-25-09969-f003:**
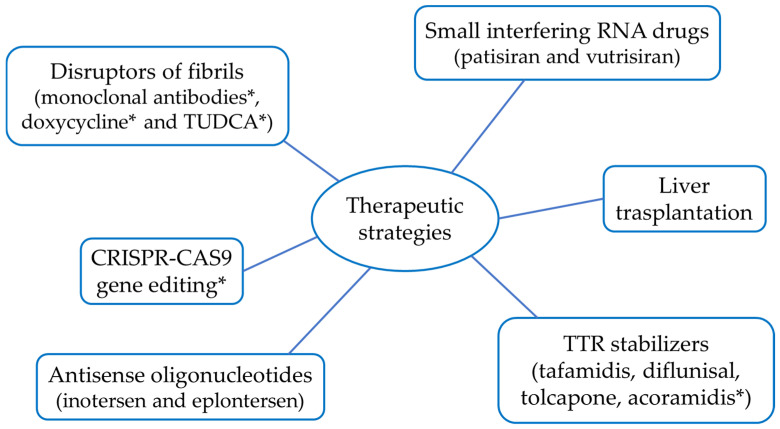
Schematic representation of the main therapeutic strategies for the treatment of ATTR. (* still under study or in clinical trials).

**Figure 4 ijms-25-09969-f004:**
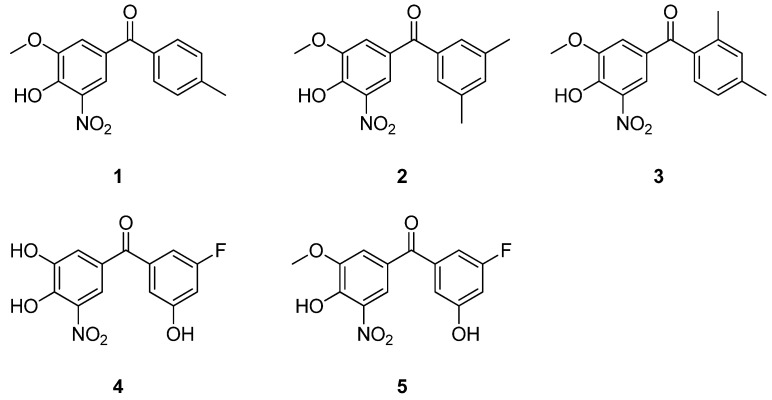
Chemical structures of compounds **1**–**5**.

**Figure 5 ijms-25-09969-f005:**
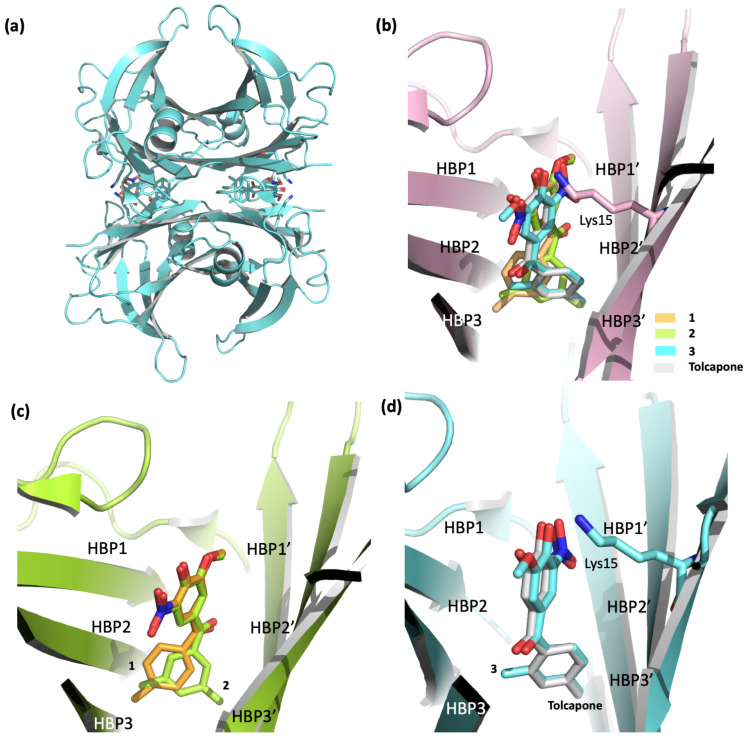
Binding modes of compounds **1**–**3** versus tolcapone (pdb id: 6SUH, 8C85, and 8C86). (**a**) Overview of TTR in complex with compound **3**. (**b**) Superposition of **1**–**3** and tolcapone. (**c**) Comparison between compounds **1** and **2**. (**d**) Detailed view of compound **3** superposed on tolcapone.

**Figure 6 ijms-25-09969-f006:**
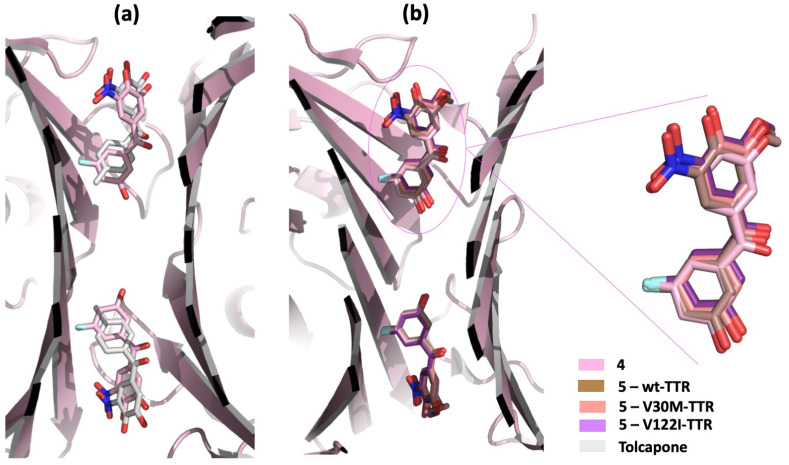
TTR in complex with compounds **4** (pdb id: 7QC5) and **5** (pdb id: 8PM9, 8PMA, and 8PMO). (**a**) Superposition between the complexes of tolcapone and **4** with wt-TTR. (**b**) Comparison of compound **5** in complex with wt-TTR, V30M-TTR, and V122I-TTR.

**Figure 7 ijms-25-09969-f007:**
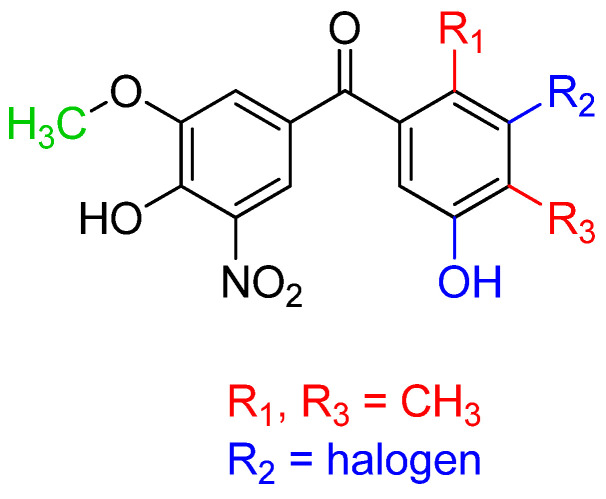
SARs of the tolcapone-derived molecules. Changes with respect to the structure of tolcapone have been highlighted in different colors based on their function and relationships.

**Figure 8 ijms-25-09969-f008:**
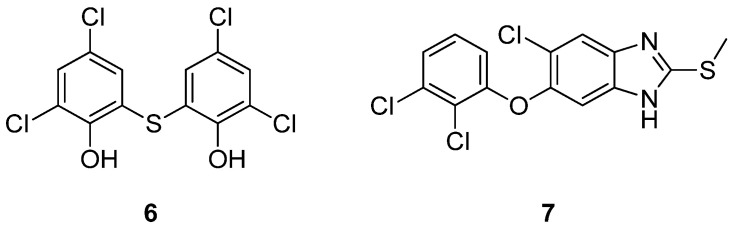
Chemical structures of compounds **6**–**7**.

**Figure 9 ijms-25-09969-f009:**
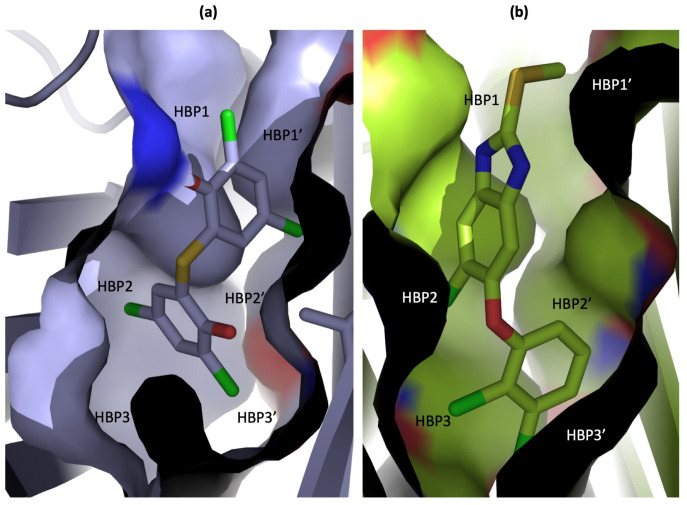
Graphical representation of T_4_BPs’ surface: wt-TTR in complex with compound **6** (pdb id: 7ERH) (**a**) and V30M-TTR in complex with compound **7** (pdb id: 7ERH) (**b**).

**Figure 10 ijms-25-09969-f010:**
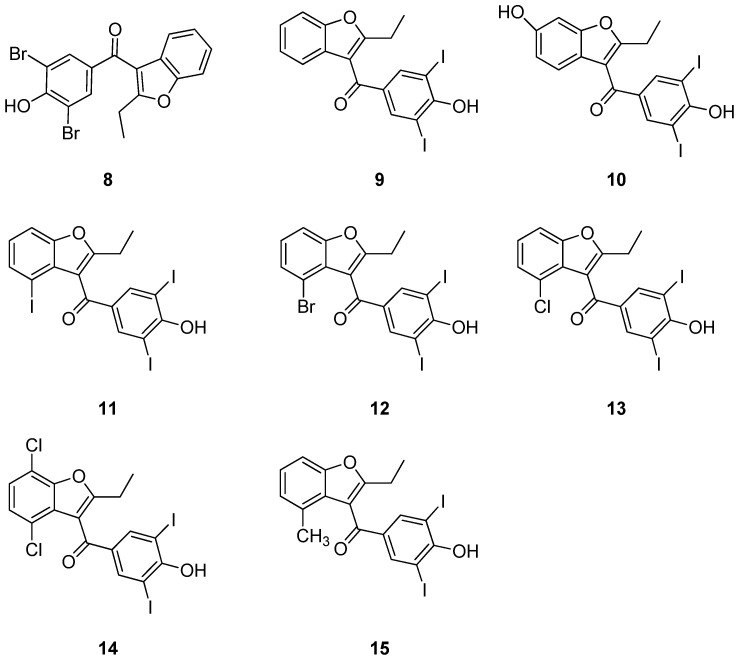
Chemical structures of compounds **8**–**15**.

**Figure 11 ijms-25-09969-f011:**
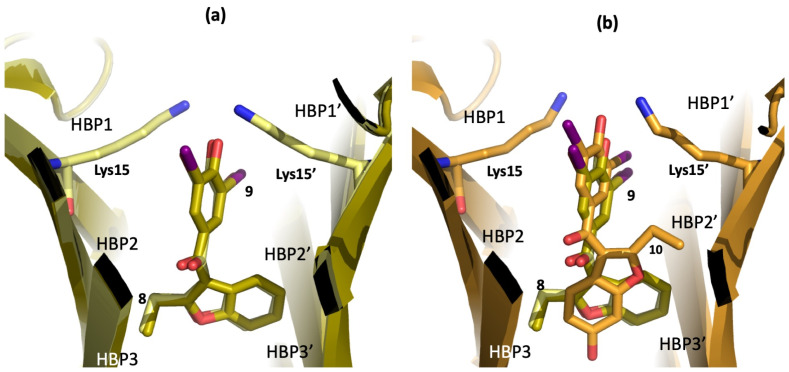
Crystal structures of compounds **8**–**10** in complex with V30M-TTR. (**a**) Superposition of **8** and **9**, pdb id 8II2 and 8II1, respectively. (**b**) Comparison between compounds **8**, **9,** and **10** (pdb id 8II4).

**Figure 12 ijms-25-09969-f012:**
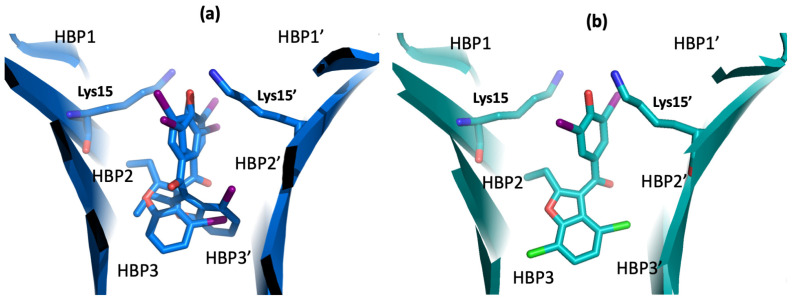
Compounds **11** and **14** in complex with V30M-TTR. (**a**) Double binding mode of **11** in the T_4_BP (pdb id 8WGS). (**b**) Pose of derivative **14** in the TTR cavity (pdb id 8WGT).

**Figure 13 ijms-25-09969-f013:**
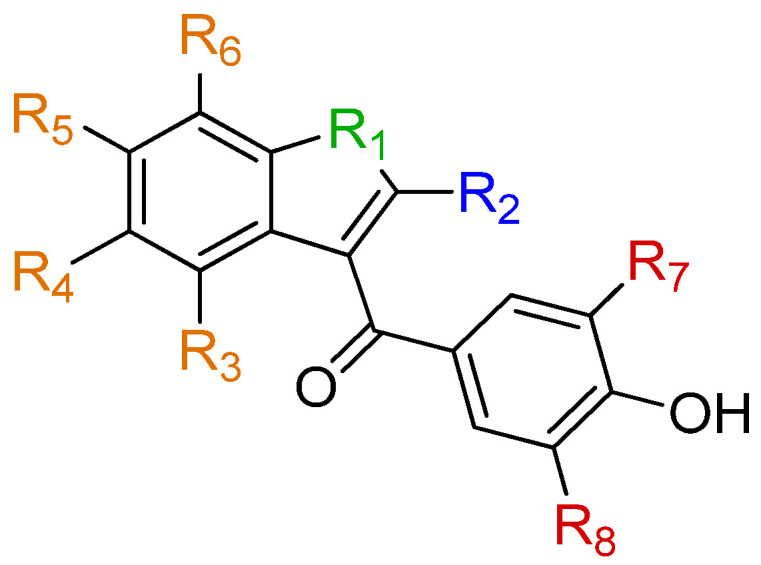
SAR for the reported scaffold, as described in [129].

**Table 1 ijms-25-09969-t001:** Binding affinity determined in terms of dissociation constant (K_d_) and enthalpy of interaction (ΔH) of the reported compounds. Abbreviations: ITC: isothermal titration calorimetry.

Structure	Technique	Values ^1,2^	Mutation	PDB Id
**1**	ITC	K_d_ = 33 ± 9 nM	wt-TTR	6SUH [102,112]
		ΔH = −8.9 ± 0.3 kcal mol^−1^		
**2**	ITC	K_d_ = 71 ± 26 nM	wt-TTR	8C85 [102]
		ΔH = −11.5 ± 1.3 kcal mol^−1^		
**3**	ITC	K_d_ = 25 ± 5 nM	wt-TTR	8C86 [102]
		ΔH = −10.5 ± 0.2 kcal mol^−1^		
**4**	ITC	K_d_ = 6.2 nM	wt-TTR	7QC5 [48]
		ΔH = −16.6 kcal mol^−1^		
**5**	ITC	K_d_ = 16 nM	wt-TTR	8PM9 [113]
		ΔH = −13.4 kcal mol^−1^		
		K_d_ = 36 nM	V30M-TTR	8PMA [113]
		ΔH = −11.0 kcal mol^−1^		
		K_d_ = 14 nM	V122I-TTR	8PMO [113]
		ΔH = −16.1 kcal mol^−1^		

^1^ The more negative the ΔH, the more selective the compound is toward TTR. The smaller the Kd, the higher the affinity [114,115]. ^2^ For some of these values, the standard deviation (SD) was not available in the reference articles.

**Table 2 ijms-25-09969-t002:** Binding affinity determined in terms of dissociation constant (K_d_) and enthalpy of interaction (ΔH) of the reported compounds.

Structure	Technique	Values ^1^	Mutation	Reference
**8**	ITC	K_d_ = 60 nM	wt-TTR	[127]
		ΔH = −9.94 kcal mol^−1^		
**11**	ITC	K_d_ = 120 ± 30 nM	V30M-TTR	[129]
**12**	ITC	K_d_ = 40 ± 19 nM	V30M-TTR	[129]
**13**	ITC	K_d_ = 66 ± 3.0 nM	V30M-TTR	[129]
**14**	ITC	K_d_ = 53 ± 16 nM	V30M-TTR	[129]
**15**	ITC	K_d_ = 42 ± 9.0 nM	V30M-TTR	[129]

^1^ For some of these values, the SD was not available in the reference articles.

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
