# Peer review of "A Snapshot of the Most Recent Transthyretin Stabilizers"

_ijms, 2024, doi:10.3390/ijms25189969_

Round 1
Reviewer 1 Report
Comments and Suggestions for Authors
The paper “Snapshot of the most recent transthyretin stabilizers” by Marotta et al. aims to describe, as stated in the title, recent new potential compounds that stabilize Transthyretin.
The paper is well written, despite its least convincing aspect lays in its dual character: in the first part it starts as a sort of review on the general treatments for transthyretin amyloidosis, whilst in part 2, much longer and well developed, enters into the details of new compounds that could act as potential new drugs against amyloidosis (which is also the point of the title).
I would suggest two modifications that could serve to improve the comprehension, in particular of readers not expert of the TTR field:
1) Better distinguish the difference between amyloidosis that arises from TTR mutations (which by the way is a “rare” disease, even if when this point is mentioned, line 133, no numbers are reported. How many people are affected? Is it rare or not?) from the disease age-related due to wild-type TTR. It is clear that the two possibly require different pharmacological strategies.
2) Outside the restrict circle of TTR experts, the words “cooperative” and “cooperativity” of the binding, which is mentioned only at pages 10 and 12, will appear quite obscure. The aspect of the negative cooperativity (that anyhow still remains a controversial point) should be introduced and explained since the beginning. In addition, this aspect is never discussed further. For example, in Table 2 the Kd values should be indicated as Kd1 (I imagine the second Kd is not available in most cases). This is particularly relevant, since the second constant can be even two order of magnitude with respect to the first one. Also the molecular reasons of the difference of binding constants is still controversial.
In any case, it is possible that the binding of one ligand is sufficient to stabilize the tetramer and avoid the formation of fibrils.
Author Response
Comment 1
The paper “Snapshot of the most recent transthyretin stabilizers” by Marotta et al. aims to describe, as stated in the title, recent new potential compounds that stabilize Transthyretin.
The paper is well written, despite its least convincing aspect lays in its dual character: in the first part it starts as a sort of review on the general treatments for transthyretin amyloidosis, whilst in part 2, much longer and well developed, enters into the details of new compounds that could act as potential new drugs against amyloidosis (which is also the point of the title).
I would suggest two modifications that could serve to improve the comprehension, in particular of readers not expert of the TTR field:
Response 1
On behalf of all the contributing authors, I would like to express our sincere appreciations of reviewer constructive comments. The comments were helpful for improving our article. According to your suggestions, we have modified our manuscript. In this revised version, changes to our manuscript were highlighted in taking mode. Point-by-point responses to the comments are listed below this letter.
Comment 2
Better distinguish the difference between amyloidosis that arises from TTR mutations (which by the way is a “rare” disease, even if when this point is mentioned, line 133, no numbers are reported. How many people are affected? Is it rare or not?) from the disease age-related due to wild-type TTR. It is clear that the two possibly require different pharmacological strategies.
Response 2
We thank the Reviewer for pointing out these unclear aspects. Regarding the difference between amyloidosis that arises from TTR mutations and the age-related disease due to wild-type TTR we dedicated several paragraphs in the sessions 1.2 and 1.3. We kindly invite the Reviewer to verify if it is clear. Hereafter is reported the corresponding extract (pages 2-3):
“This process occurs because of the intrinsic instability of the TTR protein, which can be due to either aging or single point mutations in the gene encoding it [28,29]. The first situation (namely, the aging) results in the deposition of wild-type TTR (wtTTR) fibrils, causing a condition known as “wild-type transthyretin amyloidosis” (ATTRwt) (which was previously called “senile systemic amyloidosis”) [27,29]. On the other hand, the latter causes the synthesis of less stable mutated TTR, which results in a condition known as “transthyretin amyloidosis variants” (ATTRv), also referred to as “mutant” or “hereditary transthyretin amyloidosis” [28,30]. Because of these distinct characteristics, ATTRwt is considered an age-related disease while ATTRv is hereditary [28,31].”
“As already mentioned in the previous paragraph, ATTR involves the accumulation of amyloidogenic fibrils in several different districts of the body [28]. If the depos-its are concentrated in the hearth, the patient develops a disease called “amyloidotic cardiomyopathy” (AC), while if they are located in the nerves the so-called “amyloidotic polyneuropathy” (AP) arises [31,55]. In particular, ATTR-AC results from the deposition of fibrils formed either from wtTTR or mutated TTR and, therefore, ATTR-AC can be either ATTRwt or ATTRv [55,56]”.
Comment 3
(which by the way is a “rare” disease, even if when this point is mentioned, line 133, no numbers are reported. How many people are affected? Is it rare or not?)
Response 3
According to the constructive comment made by the Reviewer, the manuscript was modified as follows (page 4):
“For several decades ATTR-AC was considered a rare disorder, while, in the last years, thanks to the great innovation in diagnostic imaging an increasing incidence has been recorded [54,57,58]. Indeed, an epidemiological study reported that in UK the incidence of ATTRwt increased from less than 3% in 1987–2009 to 14% in 2010–2015 to finally reach 25% in the years 2015-2019 [59]. However, despite the advances in diagnosis the ratio between incidence and prevalence of wtATTR in the population remains unknown [59].”.
Comment 4
Outside the restrict circle of TTR experts, the words “cooperative” and “cooperativity” of the binding, which is mentioned only at pages 10 and 12, will appear quite obscure. The aspect of the negative cooperativity (that anyhow still remains a controversial point) should be introduced and explained since the beginning. In addition, this aspect is never discussed further. For example, in Table 2 the Kd values should be indicated as Kd1 (I imagine the second Kd is not available in most cases). This is particularly relevant, since the second constant can be even two order of magnitude with respect to the first one. Also the molecular reasons of the difference of binding constants is still controversial.
In any case, it is possible that the binding of one ligand is sufficient to stabilize the tetramer and avoid the formation of fibrils.
Response 4
We thank the Reviewer for this comment and according to the suggestion the concept of cooperativity and its related terms have been now explained in the paragraph 1.2 (which are part of the introduction): “In light of this, it is clear as that molecules capable of stabilizing the TTR tetramer, and thus preventing its dissociation and subsequent misfolding, are promising therapeutic agents for the treatment for of ATTR. Usually, such stabilization occurs through their binding to the TTR binding sites and, in this regard, it is worth mentioning that they can bind in a non-cooperative or cooperative way (the latter can be further divided in positive and negative cooperativity) [45–47]. Negative cooperativity is the most frequent binding mode for TTR ligands and it means that after binding to the first TTR binding site, the affinity of the ligand for the second site decreases [48–51]. Therefore, ideally, ligands should either show positive cooperativity or non-cooperativity [48]. This is one of the challenges in the development of suitable TTR-stabilizing agents and it justifies the difficulties in finding suitable drug candidates. As a matter of fact, thisHowever, this field is still very promising, as demonstrated by the recent introduction in therapy line of research has been widely investigated over the years by the scientific community and it has led to the development of tafamidis (Vyndaqel; Pfizer), the first-in-class and the only drug approved for the treatment of ATTRs [49–51].”
Regarding the Kd values, the reference articles don’t specify whether they are Kd1 or Kd2. Therefore, we have reported them as they were depicted in the original articles.
Reviewer 2 Report
Comments and Suggestions for Authors
Marotta et al. have reported progress in the field of drug design aimed at stabilizing the TTR complex. They have highlighted the most recent studies published in the last four years, presented the latest work published by international groups, and outlined the pros and cons of each approach. The review is well written and easy to read and the paper is recommended for publication after minor changes:
- The authors do not present the mechanism of T4 binding in physiological conditions and how mutations would alter the binding. Section 1 would benefit from a detailed description of the mechanism of T4 binding, including an in-depth explanation of the residues involved and the rearrangement of the HBPs after binding.
- Figure 1: it would help to increase the size of panel C in Figure 1.
- Figure 2: please, add reference to the papers reporting the structure of the molecules.
- Line 163-164: report reference.
- Section 1.4: no mechanism of interaction has been presented for any of the stabilizers other than AG-10. The paragraph would benefit of a more comprehensive description of the stabilization performed by tafamidis, diflunisal, and tolcapone.
- Figure 4: please, add reference to the papers reporting the structure of the molecules.
- Figure 5: please, report the PDB ID structures used in the figure.
- Figure 6: please, report the PDB ID structures used in the figure.
- Figure 8: please, add reference to the papers reporting the structure of the molecules.
- Figure 10: please, add reference to the papers reporting the structure of the molecules.
Comments on the Quality of English Language
- Remove the word “noteworthy” from lines 163, 237, 242, 274, 297.
- Remove word “hopefully” from line 577.
- Remove “that of” from line 232.
- Remove “bearing this in mind” from line 253.
Author Response
Comment 1
Marotta et al. have reported progress in the field of drug design aimed at stabilizing the TTR complex. They have highlighted the most recent studies published in the last four years, presented the latest work published by international groups, and outlined the pros and cons of each approach. The review is well written and easy to read and the paper is recommended for publication after minor changes:
Response 1
On behalf of all the contributing authors, I would like to express our sincere appreciations of reviewer positive comments. In this revised version, changes to our manuscript were highlighted in taking mode. Point-by-point responses to the comments are listed below this letter.
Comment 2
The authors do not present the mechanism of T4 binding in physiological conditions and how mutations would alter the binding. Section 1 would benefit from a detailed description of the mechanism of T4 binding, including an in-depth explanation of the residues involved and the rearrangement of the HBPs after binding.
Response 2
We thank the Reviewer for this constructive comment. Following the Reviewer’s suggestion, a detailed description of T4 binding mode was added, as well as a comment regarding the effect of the mutation. In order to clarify the mode of interaction between T4 and TTR, Figure 1 was modified adding a picture of T4-TTR crystal complex. “Two molecules of T4 bind the TTR tetramer: one at the interface of monomers A/A’ and another between B/B’ subunities, with an occupancy of 0.5, respectively. Each T4BPs is characterized by three symmetric subsites which accommodate the four iodine atoms of T4 named halogen-binding pockets (HBPs); from the outer to the inner are labeled: HBP1, 1’, 2, 2’, 3 and 3’. The HBP3, is located between the side chains of Ser117, Thr119, Ala108 and Leu110, while the HBP2 is characterized by the side chains of Leu110, Ala108, Ala109 and HPB1 is contoured by Glu54 and Lys15. As depicted in Figure 1c, the crystal structure of T4 in complex with TTR (pdb id 2ROX) shows that the hormone goes deep in the BPs with the phenolic group pointing towards Ser117 and Thr119 (HBP3) while the alanyl moiety is oriented versus the entrance of the HBP1 (Glu54 and Lys15). This kind of orientation is conventionally defined forward binding mode [5]. The T4 iodine atoms established the main interactions with Ala109 and Leu110 (HBP2/3). Interestingly, a single point mutation of Leu110 with an Ala decreases the hydrophobicity of HBP3/HBP2 and thus promotes the fast dissociation of the tetramer into monomers [6].”
Comment 3
Figure 1: it would help to increase the size of panel C in Figure 1.
Response 3
As suggested, the size of panel C in Figure1 was increased, moreover, in order to address at the first Reviewer’s comment, a new picture was added.
Comment 4
Figure 2: please, add reference to the papers reporting the structure of the molecules.
Line 163-164: report reference.
Response 4
We thank the Reviewer for pointing out this unclear aspect; however, we would like to precise that each figure was made by the authors using the appropriate programs: chemdrow for chemical structures (Figure 2, 4, 7,8, 10 and 13), pymol for all the others.
Comment 5
Section 1.4: no mechanism of interaction has been presented for any of the stabilizers other than AG-10. The paragraph would benefit of a more comprehensive description of the stabilization performed by tafamidis, diflunisal, and tolcapone.
Response 5
We thank the Reviewer for the comment. From our point of view this paragraph would be an overview of all the therapeutic approaches either currently employed in therapy or under development. Appropriate references were cited in the text in order to allow the reader to deepen their study. Moreover, we would like to highlight that the mechanism of action of the stabilizers of interest (for this review) was provided in paragraph 2. However, in the case of AG10 we decided to briefly describe it in paragraph 1.4 because of its peculiarity and importance in the field.
Comment 6
Figure 4: please, add reference to the papers reporting the structure of the molecules.
Response 6
We thank the Reviewer for pointing out this unclear aspect; however, we would like to precise that each figure was made by the authors using the appropriate programs: chemdraw for chemical structures (Figure 2, 4, 7,8, 10 and 13), pymol for all the others.
Comment 7
Figure 5: please, report the PDB ID structures used in the figure.
Response 7
We thank the Reviewer for pointing out this unclear aspect; we added the pdb id in the caption. Anyway, please see also Table 1 where there is a dedicated column which reports (for each studied compound) the pdb id and the reference.
Comment 8
Figure 6: please, report the PDB ID structures used in the figure.
Response 8
We thank the Reviewer for pointing out this unclear aspect; we added the pdb id in the caption. Anyway, please see also Table 1 where there is a dedicated column which reports (for each studied compound) the pdb id and the reference.
Comment 9
Figure 8: please, add reference to the papers reporting the structure of the molecules.
Response 9
We thank the Reviewer for pointing out this unclear aspect; however, we would like to precise that each figure was made by the authors using the appropriate programs: chemdraw for chemical structures (Figure 2, 4, 7,8, 10 and 13), pymol for all the others.
Comment 10
Figure 10: please, add reference to the papers reporting the structure of the molecules.
Response 10
We thank the Reviewer for pointing out this unclear aspect; however, we would like to precise that each figure was made by the authors using the appropriate programs: chemdraw for chemical structures (Figure 2, 4, 7,8, 10 and 13), pymol for all the others.
Comments on the Quality of English Language
- Remove the word “noteworthy” from lines 163, 237, 242, 274, 297.
Reply: Done
- Remove word “hopefully” from line 577.
Reply: Done, the sentence was modified.
- Remove “that of” from line 232.
Reply: Done
- Remove “bearing this in mind” from line 253.
Reply: Done